# Lattice-Strained Bimetallic Nanocatalysts: Fundamentals of Synthesis and Structure

**DOI:** 10.3390/molecules29133062

**Published:** 2024-06-27

**Authors:** Yaowei Wang, Huibing Shi, Deming Zhao, Dongpei Zhang, Wenjuan Yan, Xin Jin

**Affiliations:** 1Shandong Chambroad Zhongcheng Clean Energy, Boxing Economic Development Zone, Boxing County, Binzhou 256500, China; wyw1119@163.com; 2Shandong Chambroad Petrochemicals, Boxing Economic Development Zone, Boxing County, Binzhou 256500, China; huibing.shi@chambroad.com (H.S.); deming.zhao@chambroad.com (D.Z.); 3State Key Laboratory of Heavy Oil Processing, College of Chemical Engineering, China University of Petroleum, No. 66 Changjiang West Road, Qingdao 266580, China; jamesjinxin@upc.edu.cn

**Keywords:** lattice strain, bimetals, catalyst, synthesis

## Abstract

Bimetallic nanostructured catalysts have shown great promise in the areas of energy, environment and magnetics. Tunable composition and electronic configurations due to lattice strain at bimetal interfaces have motivated researchers worldwide to explore them industrial applications. However, to date, the fundamentals of the synthesis of lattice-mismatched bimetallic nanocrystals are still largely uninvestigated for most supported catalyst materials. Therefore, in this work, we have conducted a detailed review of the synthesis and structural characterization of bimetallic nanocatalysts, particularly for renewable energies. In particular, the synthesis of Pt, Au and Pd bimetallic particles in a liquid phase has been critically discussed. The outcome of this review is to provide industrial insights of the rational design of cost-effective nanocatalysts for sustainable conversion technologies.

## 1. Introduction

The development of nanotechnology in recent decades has brought new opportunities for the exploration of new materials that address the issues of fossil fuel consumption and environmental pollution [1,2,3,4,5,6,7,8,9]. Metal catalysts play a central role in sustainable chemistry. Numerous studies have confirmed that the selectivity, activity and stability of bimetallic nanocatalysts are generally superior to those of their corresponding monometallic nanocatalysts in heterogeneous catalysis due to the tunable synergy between the two metallic components [10]. The catalytic synergy for bimetallic nanocatalysts is attributed to the electronic coupling between the two metals as well as geometric manipulation induced by lattice match and mismatch [11].

When the size of a substitutional atom is small, lattice strain leads to structural changes with an atomic position instead of a normal lattice position, resulting in compressive strain. In another case, surface or structural substitution with relatively larger atoms causes tensile strain [12]. Such unique structural properties have two advantages: (i) a high-indexed surface lattice facet with tunable atomic distance; (ii) a high surface–bulk ratio with controllable electronic reconfiguration [11]. By alloying two metals, the formation of heteroatom bonds and geometric effects such as strain due to the change in metal–metal bond lengths cause new electronic structures to appear in bimetallic nanoparticles [13]. So we need to precisely control the structure, size, inter-particle distance, shape, valence state, composition and multi-functionality of metal and metal alloy nanoparticles by preparation methods [14,15]. Catalytic reactions are sensitive to electronic structure and surface atomic arrangement or coordination, which can be controlled by tuning the composition and shape of nanoparticle catalysts [16]. However, several research works have also suggested that the size, shape and composition of particles could be largely irrelevant in comparison to the nature of facets in catalysis; thus, selective high- and low-index facets have been found to selectively promote adsorption, which eventually leads to an effective catalytic reaction [17]. Therefore, the concept of manipulating reduction kinetics of active metal precursors, using step-by-step solution chemistry in a controllable manner to create selective reaction sites on surfaces and the fine design of the space surrounding attached metal complexes, is known as the most popular strategy to achieve selective catalysis on surfaces [18]. Recently, atomically dispersed catalytic materials have shown significant potential in enhancing catalytic activity for certain industrially important reactions. Both experimental and theoretical efforts indicate that atomically dispersed noble metals as catalytic sites on solid supports, in the form of M-O and M-N species, are active sites or at least are involved in catalytic circles [19]. Figure 1 summarizes representative examples for various bimetallic catalysts. In particular, the synthesis of Pt-, Au- and Pd-based bimetallic particles in a liquid phase has been critically discussed.

## 2. Fundamentals of Nanoparticle Catalysts

### 2.1. Overview of Synthesis Techniques

Synthetic conditions are key for exposing catalytically active sites. Synthesis conditions may change atomic diffusion, the adsorption energy of surfactants and interaction between nanoparticles. The introduction of a second metal (bimetallic) allows greater flexibility in the design and more tunability in the controllable synthesis of various nano-architectures including nanoparticles, nanowires, nanosheets or nanotubes [10]. Most of the synthetic procedures used to prepare monometallic particles can be applied to bimetallic particle synthesis [20]. In general, three key challenges need to be addressed in the synthesis of bimetallic nanoparticles: (i) control of particle morphology, (ii) control of particle size distribution and (iii) control of nanoparticle composition [21,22]. Sankar [21] summarized four main types of mixing pattern (Figure 2) that can be identified for nanoalloys: core–shell segregated nanoalloys, subcluster segregated nanoalloys, mixed A-B nanoalloys and multishell nanoalloys. The factors influencing their formation include the strength of the bond between the two different metals, the surface energy of the two metals, relative atomic size, electron transfer between the two metals and the stabilizer–ligand used during catalyst synthesis.

### 2.2. Morphologically Controlled Synthesis

Besson et al. [23] summarized three common methods in the preparation of noble metal catalysts with the catalytic oxidation of various polyols: the impregnation method, the deposition–precipitation method and the sol-immobilization method. Sol-immobilization is a widely used method to prepare gold catalysts [24]. Papers in the literature have summarized eight general synthetic methods [25,26]: molecular beams, chemical reduction, thermal decomposition of transition metal complexes, ion implantation, electrochemical synthesis, radiolysis, sonochemical synthesis and biosynthesis. Several common preparation methods of supported catalysts will be introduced in the following.

#### 2.2.1. Encapsulation Method by the Confinement Effect

Traditional nanocatalysts directly exposing metal active sites will lead to agglomeration and sintering, which will reduce the cycle stability of the catalyst [27]. Metal nanoparticles can be encapsulated in zeolite pores to improve the stability of the catalyst. Besides this, its unique pore structure can provide selective catalysis and improve the selectivity of the target product. Synthesis methods of confined catalysts include the ion exchange method, the impregnation method and the in situ encapsulation method. Among these, the ion exchange and impregnation methods have strict requirements for the properties of raw materials for synthesis. The in situ encapsulation method can synchronize the encapsulation of metal nanoparticles with the formation of zeolite crystals and achieve the accurate anchoring of metal clusters by stabilizing ligands or precursors (Figure 3a). Several noble metal clusters (such as Pt, Pd, Ru and Rh) have been successfully encapsulated in zeolite crystals using ligand stabilization methods and have shown good catalytic activity and selectivity in catalytic reactions [2,28].

#### 2.2.2. Electrochemical Deposition Method

During the electrochemical deposition method, a mixed solution of substrate and metal precursor salt is reduced and deposited by electrochemical methods such as cyclic voltammetry or square wave scanning to the diffusion layer, the electrolyte membrane, or the interface between the diffusion layer and the electrolyte membrane to prepare the required nanomaterials (Figure 3b). Liu et al. [29] reported an example of preparing porous PtCo nanowires by electrochemical deposition. They first deposited Pt and Co nanoparticles on porous anodic aluminum oxide (AAO) membranes in H_3_BO_3_ solution and then removed the AAO membranes in an acidic solution to obtain porous PtCo nanowires. Compared with the most advanced Pt/C and PtCo/C catalysts, the prepared porous PtCo alloy nanowires had significantly enhanced electrocatalytic activity for methanol oxidation. Because parameters such as voltage, deposition time, temperature and electric flux are easy to control, the electrochemical deposition method can achieve precise control of the morphology and size of nanoparticles.

#### 2.2.3. Liquid-Phase Chemical Reduction Method

Nanocrystals over supported catalysts can also be obtained from a metal precursor with a reducing agent. This method can precisely control crystal morphology and size by controlling the type, concentration or feeding sequence of reactants, reaction temperature, reaction time, etc. Therefore, it is the main method of industrial production. Lu and coworkers reviewed research progress of the size and shape control of magnetic nanoparticles [27]. Yang found that compared with physical methods and gas-phase strategies, hydrothermal routes are much more easily controlled and can produce nanoarrays in a designed structure and morphology for super-capacitors and catalysts [28]. Several parameters, such as temperature, reaction time, facet bias of capping agents and reduction potential of any of the involved agents, significantly influence the eventual morphology of nanocrystals [10]. Compared with other approaches, wet chemical synthesis methods have been more popular with good potential for implementing environmentally friendly production routes. Green solvents and weak reductive agents are believed to pose less environmental impact during preparation processes [10].

#### 2.2.4. Ultrasound-/Microwave-Assisted Reduction Method

Ultrasound-assisted reduction is a method in which a shock wave can form huge pressure on the surface and channel of a catalyst, discharging gas out of the hole and influxing an active component into the channel, which allows the catalyst component to be distributed in the channel easily. In Liu’s research, hydrogen radicals are excited from plentiful hydroxyls under the action of ultrasound during preparation [30]. Ru-supported Ni-FeLDH catalysts prepared by the ultrasound-assisted method performed a conversion of 100% in N-ethylcarbazole (NEC) hydrogenation and retained excellent catalytic stability in the reaction.

In addition, another method, microwave-assisted synthesis, is where a uniformly mixed precursor reaches a certain high temperature through the absorption of microwave energy, thereby triggering high dispersal. Ni et al. reported the solvent-free microwave-assisted synthesis of supported ruthenium nanocatalysts uniformly supported on non-functional carbon nanotubes (CNT) using Ru_3_(CO)_12_ as a precursor, showing 80% conversion and 72% selectivity for cinnamaldehyde hydrogenation [31]. Microwave-assisted synthesis is a simple and promising alternative technology for preparing highly dispersed metal-based catalysts.

#### 2.2.5. Microemulsion Method

The microemulsion method usually refers to two immiscible solvents forming a microemulsion, in which metal ions are nucleated, grown and heat-treated under the action of a surfactant to obtain nanoparticles with a narrow size distribution [32,33,34,35,36,37]. The microemulsion method is an improving method to prepare monodispersed nanoparticles because of the controllable particle size by varying the size of the reversed micelles. The micelle size of the microemulsion and the exchange rate between micelles are important parameters to determine the properties of the nanoparticles [32].

PtCo nanoparticles of uniform size (about 3–4 nm) were synthesized by the simultaneous reduction of H_2_PtCl_6_ and Co(NO_3_)_2_ with NaBH_4_ using a W/O microemulsion (water/16.5% polyethylene glycol dodecyl ether/n-heptane) [34]. The cubic phase was formed. PtCo nanoparticles present superparamagnetic behavior and exhibit highly reactive activity to oxalic acid oxidation in H_2_SO_4_. Anil et al. prepared microemulsions with different droplet diameters by changing the molar ratio of water to dioctyl sodium sulfosuccinate (AOT) and explored the formation mechanism of nanoparticles in microemulsions in a comprehensive experimental and simulation study [35].

Moreover, nanoparticle catalysts can be used as promoters that are applicable in both homogeneous and heterogeneous phases [38,39]. Nanoparticle catalysis in a homogeneous medium, often referred to as “soluble” NP catalysis, is a classic topic in the field of green chemistry [40]. Initially, the nanoparticles obtained are usually dispersible only in organic solvents, as synthesis procedures usually involve the addition of large amounts of ligands to stabilize particles. Both ligands and solvent are critical for all stages of particle growth and crystallization, impacting composition, size, shape and surface properties of the nanocrystals. Biphasic systems were later developed to tune growth kinetics in a liquid medium [41]. But organic solvents are still the most popular approach to tailor the nanostructures of particles or clusters. For example, oleylamine (OAm) is usually used as a solvent, surfactant and reducing agent, forming metal-OAm complexes at intermediate temperatures (T~250 °C) to control the decomposition rate for generating nanoparticles [42].

## 3. Synthesis and Properties of Bimetallic Nanoparticles

### 3.1. Au-Based Bimetallic Nanoparticles

Wet chemical methods are widely used to prepare various Au-based bimetallic catalysts. Recent efforts have been focused on preparing AuPd [43,44,45,46,47], AuPt [48], AuAg [49,50,51] and AuCu [52,53,54] nanoparticles. Typical methods involved in wet chemical approaches include precipitation in the presence of alkaline agents (e.g., NaOH [50]) and in situ deposition induced by introducing reductive agents such as NaBH_4_ [44,45,46,55], methyl ammonia borane and oleic acid. Structural directing agents such as rGO [46,52], PVP [55,56] and hydroxyethyl starch-poly can be added to control the size and shape of nanoparticles. Typical supports for Au-based bimetallic nanoparticles include graphite, carbon, FTO glass [43,48], TiO_2_ [53], ZnO [54] and rGO [46,52]. Other approaches, including thermal annealing and plasma reduction, can also be combined to further tune the shape or oxidation states of particles. Au-based bimetallic catalysts are summarized in Table 1.

For AuPd catalysts prepared by thermal annealing, XRD and HRTEM patterns confirm the formation of alloy AuPd particles rather than bulk metals with a miscibility gap. Without a special preparation method, Au and Pd metals can form alloys in almost any proportion while maintaining a face-centered cubic (*fcc*) structure [44,45]. Fast Fourier transform (FFT) analysis on both sides of the twin boundary shows that most of the particles have decahedral multiple twin structures, which maintain an *fcc* structure. The typical lattice spacing of the measured (111) plane is between Au (2.35 Å) and Pd (2.25 Å) [44]. Besides this, the lattice spacing of a Pd/Au NP alloy (111) is 0.229 nm, which is in the range of 0.234 nm and 0.223 nm, corresponding to the interplanar spacing of *fcc* (111) Au and Pd, respectively [45,57]. Because of their superior structural properties, many materials are potentially suitable as excellent substrates to immobilize and to disperse AuPd particles for liquid-phase oxidation, such as g-C_3_N_4_ with a large surface area and special p-bonded planar structure [45]. rGO is another type of remarkable support for AuPd catalysts. The corresponding size distribution of AuPd/rGO reveals that their size ranges from 1.6 to 2.8 nm, forming optimized core/shell structures [46]. A core–shell structure is also formed on the outer surface of activated carbon cloth (ACC), with large particles consisting of a gold-rich core covered by small Pd particles with a size of mostly less than 10 nm [58]. Besides this, AuPd alloy also shows a nanoporous dendritic morphology, the growth of which is driven by localized surface plasmon resonance [43].

**Table 1 molecules-29-03062-t001:** Preparation and morphology of Au-based nanoparticles.

Name	Preparation	Morphology	Ref
AuPd/HOPG *	Ar, HOPG, UHV, 300 °C	“Au core–Pd shell”	[47]
AuPd/g-C_3_N_4_-N *	one-pot deposition reduction method, melamine, sodium borohydride	the lattice edge distance is 0.229 nm, in the range of 0.234 nm and 0.223 nm, respectively	[45]
PdCore–AuShell/rGO	two-step protocol, rGO *, MeAB, Ag	spherical, varying in size from 1.6 to 2.8 nm	[46]
AuPd/activated carbon cloth	impregnation method, nitric acid, activated carbon cloth, PdCl_2_, HAuCl_4_		[58]
AuPt/FTO glass	a new process for efficiently synthesizing supported metal (or metal oxide) NPs using dry plasma reduction at near room temperature under atmospheric	AuPtBNP sample shows a finer morphology, where Au sputtered particles produce uniform coverage FTO tight junctions of glass	[48]
AuAg/TiO_2_	sequential deposition method, NaOH, urea, hydrogen	530 nm	[49]
AuAg/HES	graft copolymer reduced method, acrylamide-co-acrylic acid, NaOH, potassium bisulfate		[50]
AuCu/rGO	deposition–precipitation method, NaBH_4_, natural graphite powder, HAuCl_4_·3H_2_O, Cu(NO_3_)_2_·3H_2_O	no agglomeration, average size 15 ± 1 nm, surface spacing 0.22 nm	[52]
AuCu-ZnO-Gr	CuAu-ZnO-Gr of two-step synthesis, OLA *	ellipsoid, size 18.0 ± 2.0 nm	[54]
AuRh	hydrogen sacrificial reduction method, alcohol reduction method, sodium borohydride reduction method, PVP *, MNPs, Rh/Au	Rh (0.22 nm), Au (0.24 nm)	[55]

* HOPG: Highly oriented pyrolytic graphite. OLA: oleinic acid. PVP: polyvinyl pyrrolidone. rGO: reduced graphene oxide. g-C_3_N_4_-N: graphitic carbon nitride.

AuPt catalysts have also been widely studied. Au ions whose redox potential is higher than that of Pt ions should be more easily reduced than Pt ions. Thus, Au atoms are first gathered to form many tiny clusters which act as seeds for the further aggregation of Pt atoms [59,60]. Jin’s group also found that subtle differences in electronegativity between Pt and Au may also essentially affect the final morphology [60,61]. A large amount of Pt and Au nanoparticles formed instantaneously under the strong reducibility of NaBH_4_. Besides this, TiO_2_ provides an electron transfer channel from Au^3+^ to Pt^4+^ (Au−Pt interaction), which contributes to the co-reduction of Au and Pt ions, providing a possibility for AuPt alloys. The anisotropic growth of AuPt clusters overcomes strong interfacial forces, leading to the formation of twin boundaries within the bimetallic particles.

AuAg catalysts are one of the Au-based systems that show surprising performance in terms of stability and activity, because of the interaction of Au-Ag rather than a simple mixture of nanoparticles [49,50,51]. Reducibility of Au and Ag during thermal activation was studied by UV–Vis (Figure 4a) [50]. Ag-Au nanoparticles prepared by grafting copolymer produced a single absorption peak at 456 nm that is different from Ag or Au single-metal nanoparticles under the same experimental conditions, indicating that it is not a simple mixture of single-metal nanoparticles. The CO adsorption peak of 2055 cm^−1^ over Au-Ag/TiO_2_ catalysts (Figure 4b) perfectly corresponds to the CO-Ag^0^ species, attesting that Au-Ag atoms have stronger affinity toward O_2_ and can dissociate it [62]. The optimal activation temperature of 550 °C is sought on a Au-Ag/TiO_2_ catalyst, based on the particle size effect (negative) and bimetallic interaction effect (positive) [49].

AuCu particles supported on rGO show insignificant agglomeration with an average size of 15 ± 1 nm (Figure 5) [52]. Characterization demonstrates the polycrystalline nature of synthesized nanoparticles. EDS elemental mapping patterns further revealed that Au and Cu elements are uniformly distributed, fully indicating the formation of an Au–Cu alloy without phase segregation. Mechanistic studies for AuCu/ZnO catalyst formation suggest that bimetallic alloy CuAu NPs can cause more photo-generated electrons to transfer to the conduction band of ZnO under simulated sunlight irradiation. Therefore, ZnO can absorb a small part of ultraviolet light in simulated sunlight to generate conduction band electrons (e^−^) and valence band holes (h^+^) [54]. As a consequence, electrons that come from CuAu and ZnO accumulate together on the conduction band of ZnO, and these electrons are trapped by dissolved O_2_ molecules in water as superoxide radical anions and hydroxyl radicals. The oxidized positively charged CuAu NPs and the valence band holes (h^+^) capture e^-^ from water and colorant molecules to neutralize the positive charge.

The surface plasmon resonance (SPR) band for AuCu/TiO_2_ catalysts is shown in Figure 6. Au was red-shifted to 545 nm with the addition of Cu^2+^ on its surface accompanied with a change in color from pink to light red. With the progressive addition of Cu^2+^, the intensity of the peak at 560–580 nm increased due to a change in shell thickness as well as its composition. A significant change in color and absorption peak was found with an increasing amount of Cu^3+^. This result indicated the coexistence of two elements (Cu and Au) in one composite, and Au (NS) was entirely encapsulated by Cu (NS), giving rise to a Au core–Cu shell structure.

Other bimetallic Au-based catalysts such as AuRh also show good performance for oxidation. Experimental studies show an increase in the average size of the prepared particles with an increase in Au content, indicating that the Au component was indeed reduced on the surface of the Rh particles. The as-prepared AuRh particles possessed high catalytic activity for H_2_O_2_ decomposition. The catalytic activity of the prepared AuRh was closely dependent on its composition. The activity of AuRh nanoparticles with an average size of 2.7 nm was about 3.6 times higher than that of pure Au and Rh monometallic catalysts. Density functional theory (DFT) calculation showed that charged Rh and Au atoms formed via electronic charge transfer effects could be responsible for the high catalytic activity.

### 3.2. Pt-Based Bimetallic Nanoparticles

In the case of a bimetallic PtFe system, where Pt has a larger lattice distance than Fe, Fe^3+^ contributes to both galvanic displacement and a lattice-mismatched template for atomic deposition of Pt species. The combinatory effect eventually assists the formation of pyramid-shaped bimetallic clusters with high-surface-index morphologies (Figure 7) [56]. Different synthetic approaches result from different morphologies.

A PtMn system, where the size of the Mn crystal cell is much larger than Pt, however, is completely different from a PtFe system. Ordered structures of PtMn cannot occur due to strong lattice strain at the Pt–Mn interface. Thermal annealing can produce strain-relaxed ordered tetragonal structures [63,64,65,66,67,68,69]. Recent experimental work on lattice-strain-induced structure distortion reveals that a slight increase in Mn content in Pt_1_Mn_x_ clusters (x: 0.5–2.0) causes Oswald ripening of nanoparticles during self-assembly [63]. The difference in PtFe and PtMn systems suggests that Pt compressive and stretching growth due to lattice strain might follow different mechanisms. In particular, as Mn/Pt content increases from 0.5 to 2.0, bimetallic PtMn clusters evolve from nano-buds to asymmetric cauliflower shapes. Such dramatic cluster morphology reveals that a large mismatch between Pt and Mn with a faster reduction rate of Pt causes simultaneous growth of multiple bud-arms at initial stages. More Mn content stimulates the anisotropic growth of each arm; thus, flower-shaped clusters are achieved (Figure 8). The asymmetric morphology further confirms the hypothesis that the nuclei formed initially immobilize on a solid surface; thus, the whole growing process occurs towards a solvent medium.

The wet synthesis of PtCo as well as PtFe nanocrystals follows a different story due to completely different crystallinity between Pt and Co/Fe metals [70,71]. Sun’s group discovered surprising lattice-dependent activity behavior on PtCo and PtFe nanowires [72]. A wire-shaped morphology can be obtained by injecting Co and Fe carbonyl into Pt(acac)_2_ solution in a 1-octadecene medium. Due to the strong lattice strain between Pt and Co/Fe metals, disordered *fcc* structures are obtained for as-prepared nanowires [73]. By thermal treatment, however, *fcc* evolve to ordered fct structures and exhibit enhanced surface activity for ORR applications.

Compared with PtFe, PtCo and PtMn systems, other bimetallic Pt-based combinations such as PtNi and PtCu also exhibit strain-induced morphology-controlled growth [74]. But it is important to mention that lattice strain in the latter systems is favorable for the selected growth of one specific facet rather than stimulating disordered structures. For example, Ni-assisted Pt nanoparticle growth kinetically accelerates a Pt (100) surface; thus Pt (111) with octahedral crystals is dominant in final samples [75,76,77,78]. Similarly, due to a minor lattice mismatch between Pt and Cu, lattice strain can actually act as the driving force for *fcc* architecture formation for the generation of (111), (110) and (100) facets [79,80,81,82]. For example, Jiang and Xie [83] synthesized rhombic dodecahedral PtCu_3_ alloy nanocrystals with a high-energy surface (110) using *n*-butylamine as a surface regulator. From a crystallographic point of view, the formation of an excavated polyhedron is very difficult due to surface energy minimization and close lattice match between Pt and Cu. It is believed that the strong adsorption of *n*-butylamine on a PtCu_3_ (110) surface causes strain reconstruction at the Pt and Cu interface, leading to the growth of a thermodynamically favorable (111) surface while a (110) surface is maintained [83,84]. In other words, the competitive adsorption of amines and minor lattice mismatch might be the intrinsic driving force for the eventual formation of such high-indexed structures.

The classic galvanic displacement technique has also been investigated, coupled with the selective deposition of Cu^0^ species on an exposed high-energy surface. In particular, the introduction of Br^-^ significantly slows the reduction kinetics of Pt^2+^ in liquid medium (Table 2, entry 2); thus cubic structures with (100) facets can be generated with the aid of reductive PVP species [85]. In the second stage, the galvanic displacement of Cu^0^ by Pt^2+^ etches low-indexed surfaces, which is followed by a final stage of reduction in Cu^2+^ under hydrothermal conditions, allowing eventual surface reconstruction or strain relaxation to achieve concave structures. Similarly, kinetic control is also used to assist the formation of high-indexed structures for PtCu nanoclusters.

Temperature also plays an important role in the morphology of Pt_3_Ni nanocrystals. Yang’s group prepared truncated-octahedral Pt_3_Ni catalysts with (111) as the dominant facet [86]. The length of carbon chains in an amine agent is critical for the formation of a truncated-octahedral structure. Shorter alkane chain amines favor the formation of a (111) surface, suggesting that kinetic rate control to release lattice strain in Pt and Ni system is important. Fang’s team prepared Pt_3_Ni nanooctahedra using W as the surface modifier [87]. Since it is difficult to slow nucleation time period under solvothermal conditions, W was employed to self-provide stable sources of Pt clusters in the growth stage. The displacement reaction between Pt^2+^ and W^0^ controls the thermodynamic growth of Pt_3_Ni crystals by the difference in the surface energy on each crystallographic face; thus, finely tuned octahedral structures can be obtained. A PtNi octahedral with defects at the corner was prepared using polyol reduction in the presence of poly (diallyldimethylammonium chloride). Element distribution analysis reveals that Pt is richer in corners and edges where lattice strain is strong [88,89,90]. Debe’s team reported novel Pt_3_Ni_7_ nanocrystals with *fcc* lattice parameters of 0.37 nm and 7.5 nm in size. However, such a composition is found to suffer from Ni leaching due to unstable structures above Pt_3_Ni composition [91].

**Table 2 molecules-29-03062-t002:** Preparation and morphology of Pt-based nanoparticles.

Name	Preparation	Morphology	Ref
Pt-Fe/Al_2_O_3_	H_2_PtCl_6_·6H_2_O, cetyltrimethylammonium bromide, butanol, cyclohexane, N_2_H_4_ H_2_O, Al_2_O_3_, ethanol, H_2_PtCl_6_, FeCl_3_ 80 °C	Polycrystalline structure with high crystallinity	[32]
PtCu NPs	Platinum (IV) chloride, copper (II) acetylacetonate, oleylamine, polyvinylpyrrolidone, ethylene glycol 200 °C	Spherical, mean particle size (7.0 ± 0.7) nm	[85]
PtCu NWs	Platinum (II) acetylacetonate, cupricchloride, dihydrate, hexacarbonyltungsten, C_6_H_12_O_6_, dodecyl trimethyl ammonium bromide, oleylamine, 1-octadecene, C_6_H_12_, CH_3_COOH, CH_3_CH_2_OH, CH_3_OH 170 °C	Nanowire structure	[92]
PtCu/TiO_2_	Pt, Cu, Ar, 500 °C	Nanotube structure	[93]
PtNi/HSNs	H_2_PtCl_6_·6H_2_O, Ni(CH_3_COO)_2_·4H_2_O, oleylamine, cetyltrimethyl ammonium chloride, H_2_SO_4_ 160 °C	hierarchical framework of multilayer structure, uniform octahedral shape, narrow particle size distribution, mean particle size nm 79.8	[94]
Pt-Ni/CeO_2_	Ce(NO_3_)_3_·6H_2_O, NaOH, NiCl_2_, H2PtCl_6_, H_2_/N_2_ 600 °C	Average particle size 2 nm	[95]
Pt–Ni/ZnO-rod	Zn(NO_3_)_2_·6H_2_O, NaOH, Pt, Pt-acetylacetonate, acetone, Ni(NO_3_)_2_·6H_2_O, ethanol, Ni, H_2_ 500 °C	Nano-rod structure 3–14 nm	[96]
Pt-Co/TiO_2_	Pt(C_5_H_7_O_2_)_2_, CoC_22_H_14_O_4_, Ti(OCH_2_CH_2_CH_2_CH_3_)_4_ 500 °C	Large spherical particles	[97]

Different from alloy structures, core–shell morphologies have radial and compressive lattice strain due to lattice mismatch between core and shell materials. Johnson and coworkers predicted trends of more than 132 alloy and core–shell structures for late transition metal systems using DFT [98]. They systematically explored possible segregation energies to determine surface migration trends for binary systems because segregation energy provides a quantitative assessment of segregation behaviors in alloy systems. In general, the core–shell preference from a segregation energy perspective is largely determined by two independent parameters, namely cohesive energy and Wigner–Seitz radius, where the former is considered to be the primary factor. In particular, elements with relatively stronger cohesive energy go to the core part, while metals from the same group with a smaller Wigner–Seitz radius determine core–shell morphologies. Chemical environment also plays a critical role in the formation of core–shell structures. Tsung and coworkers observed the migration of Au from the core to a Pd-rich thin shell due to differences in surface and cohesive energies [99]. Pd has a relatively larger cohesive energy compared with Au. It is therefore more favorable for Pd to migrate to the core part rather than being under-coordinated on the edge [100]. The cohesive energy effect is the main driving force for the formation of PdAu core–shell structures.

## 4. Conclusions and Outlooks

Based on critical discussion of Au- and Pt-based bimetallic nanocatalysts, it is clear that synthetic conditions are key for controllable morphologies. Temperature, pressure, aging time and concentration of template agents are among the most important factors that contribute to eventual morphologies. However, the following issues remain in current studies:**(i)** Removal of template agents. Despite beautiful and controllable morphologies owing to the addition of template agents, they are detrimental for catalysis, as residual species of polymers block active sites for surface reactions. Therefore, future efforts should be paid to develop efficient methods to effectively remove template agents, rather than merely focusing on morphological control.**(ii)** Scale-up synthesis is still missing. Although existing studies have revealed the unique kinetics of the formation of bimetallic nanoparticles, experimental studies on scale-up synthesis are still missing. This might mislead young researchers by an ill-defined yield of nanoparticles per precursor added, as such information is not available in most of the literature.

Nevertheless, more quantitative studies on synthesis, catalysis, plasmonics and magnetic aspects are useful for the further development of nanotechnologies for bimetallic nanomaterials.

## Figures and Tables

**Figure 1 molecules-29-03062-f001:**
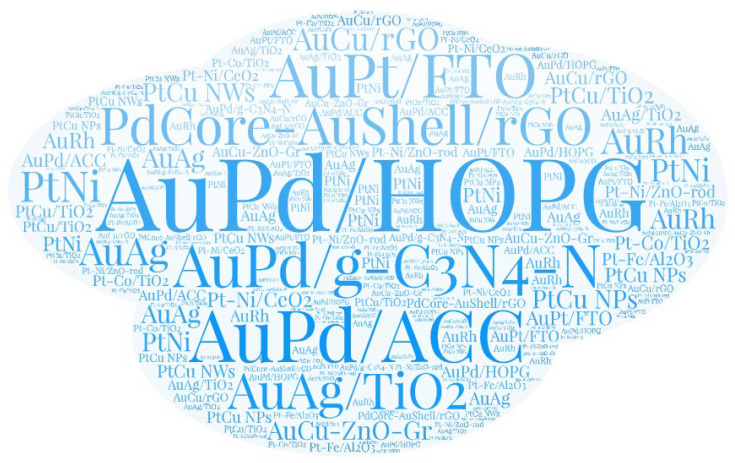
Au and Pt composites are the most promising catalysts for the future chemical industry.

**Figure 2 molecules-29-03062-f002:**
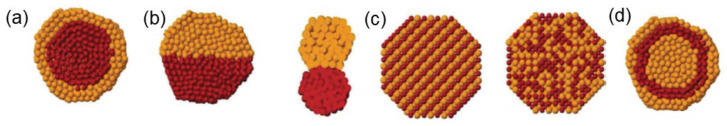
Schematic representation of some possible mixing patterns in bimetallic systems: (**a**) core–shell alloys, (**b**) sub-cluster segregated alloys, (**c**) ordered and random homogeneous alloys, and (**d**) multishell alloys. Reproduced with permission from Ref. [21]. Copyright 2012, Royal Society of Chemistry.

**Figure 3 molecules-29-03062-f003:**
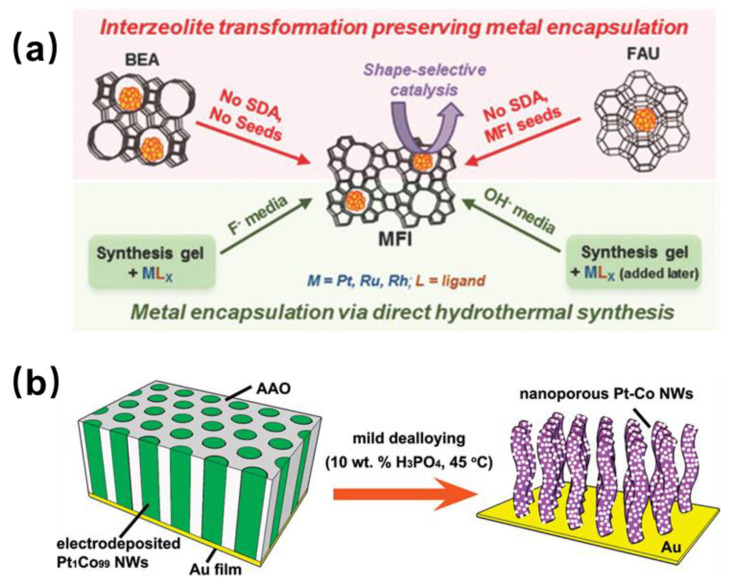
(**a**) Schematic of different methods for encapsulating metal within MFI zeolite. Reproduced with permission from Ref. [27]. Copyright 2018, Wiley. (**b**) Schematic illustration of the fabrication process of nanoporous Pt–Co alloy nanowires. Reproduced with permission from Ref. [29]. Copyright 2009, American Chemical Society.

**Figure 4 molecules-29-03062-f004:**
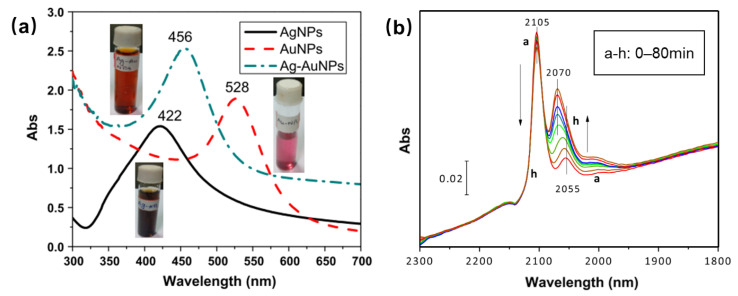
(**a**) UV–VIS spectra of Au, Ag and Ag-Au bimetallic nanocomposites synthesized by the graft copolymer HES-g-poly. Reproduced with permission from Ref. [50]. Copyright 2017, Elsevier. (**b**) DRIFT spectra of Au–Ag/TiO_2_ after in situ reduction and CO adsorption. Evolution of the spectra with CO contact time from 0 to 80 min (a–h). Reproduced with permission from Ref. [49]. Copyright 2015, Elsevier.

**Figure 5 molecules-29-03062-f005:**
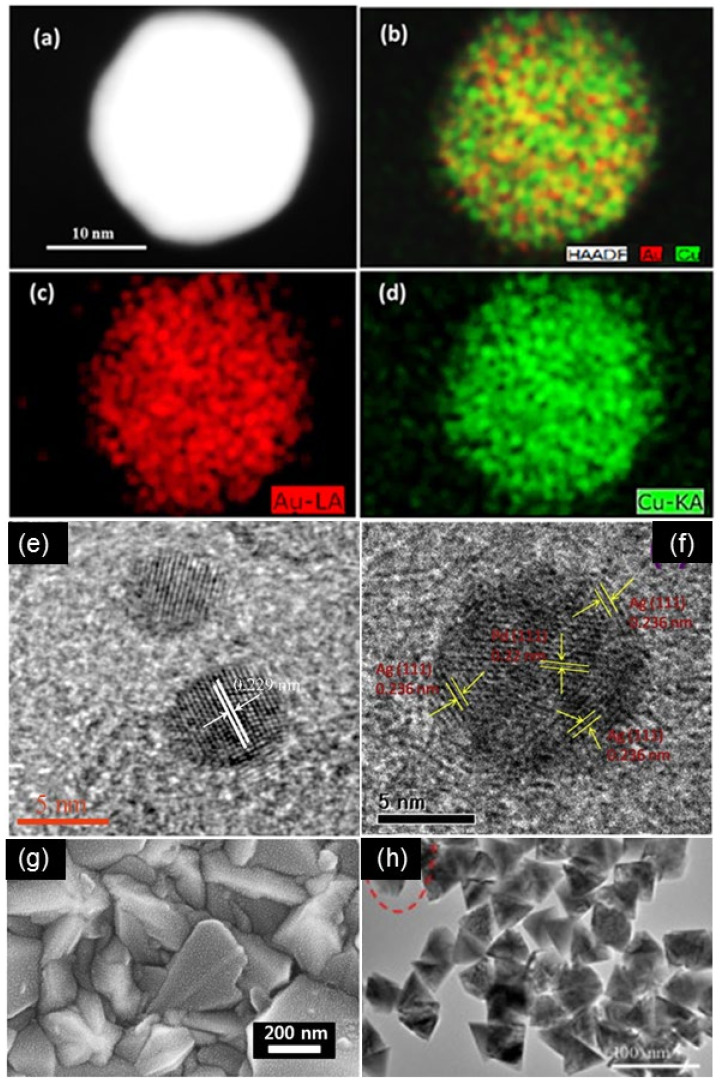
(**a**) HAADF-STEM image of Au_3_-Cu/rGO catalyst; (**b**–**d**) EDS mapping of Au_3_-Cu/rGO catalyst. Reproduced with permission from Ref. [52] Copyright 2017, Elsevier; (**e**) typical HRTEM images of Pd/Au alloy NPs in the as-obtained Pd/Au@g-C_3_N_4_-N(1:1). Reproduced with permission from Ref. [45] Copyright 2017, Elsevier; (**f**) HRTEM image of Pd@Ag/RGO. Reproduced with permission from Ref. [46]. Copyright 2018, Elsevier; (**g**) HRSEM images of AuPt-BNP/FTO glass. Reproduced with permission from Ref. [48]. Copyright 2015, Elsevier; (**h**) TEM image of ZnO nanopyramids. Reproduced with permission from Ref. [54]. Copyright 2015, Elsevier.

**Figure 6 molecules-29-03062-f006:**
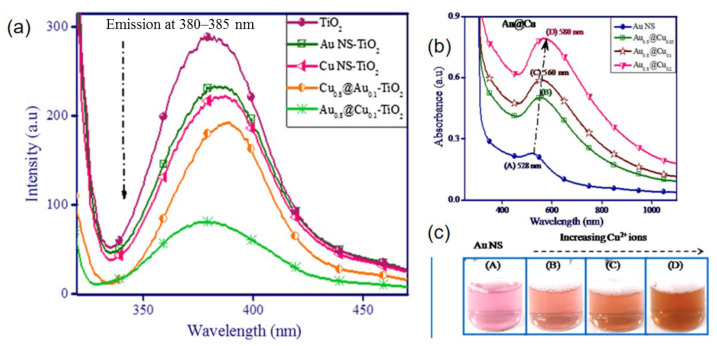
(**a**) Emission spectra of monometallic and bimetallic nanocomposites modified TiO_2_. (**b**) Effect of different amounts of CuSO_4_ (0.01 M) deposition onto Au nanospheres for variation in the surface plasmon band, and (**c**) their respective color changes (increasing Cu^2+^ ions from A to D). Reproduced with permission from Ref. [53]. Copyright 2017, Elsevier.

**Figure 7 molecules-29-03062-f007:**
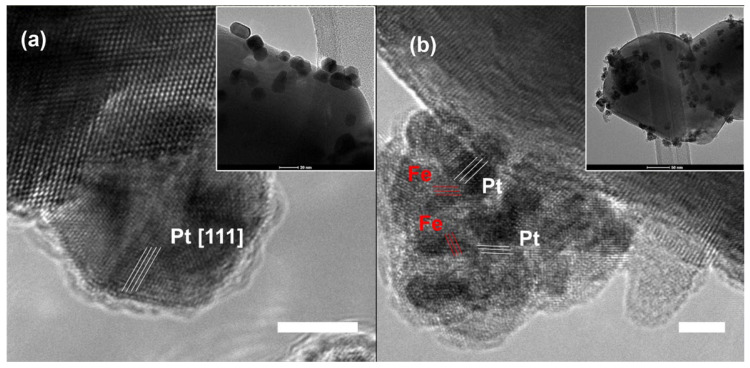
Lattice-mismatched anisotropic growth of PtFe nanoclusters. HR-TEM images of (**a**) Pt and (**b**) PtFe(1) with insets at lower magnification. White bars indicate 5 nm. Reproduced with permission from Ref. [56]. Copyright 2016, Elsevier.

**Figure 8 molecules-29-03062-f008:**
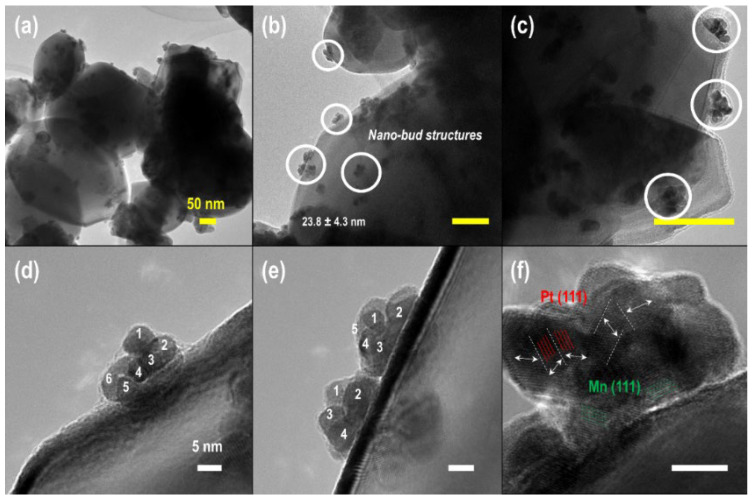
TEM images of bimetallic PtMn catalysts. (**a**–**c**) novel nano-bud shaped bimetallic clusters, (**d**,**e**) 4 to 8 buds on the PtMn clusters. Anisotropic growth orienting from the surface plane of ordered Pt octahedral or cubic structures. (**f**) Lattice-strain-induced distorted bimetallic PtMn nanocatalysts. Reproduced with permission from Ref. [63]. Copyright 2017, Elsevier.

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
