# Peer review of "Lattice-Strained Bimetallic Nanocatalysts: Fundamentals of Synthesis and Structure"

_molecules, 2024, doi:10.3390/molecules29133062_

Round 1

Reviewer 1 Report

Comments and Suggestions for Authors

1.     Please provide more relevant references in recent five years. There are too many old references which are difficult to express the latest breakthrough in this field.

2.     Please unify line spacing in this work especially in Line 181-200.

3.     The explanation of rGO in Line 250 is not correct. And the corresponding information in Table 1 is not correct, please check these references. The Preparation in Table 1 is hard to distinguish, please divide the preparation information separately.

4.     Actually, the synthesis law of these bimetallic catalysts should be summarized and analyzed. While the authors mainly focused on the only one reference.

Reviewer 2 Report

Comments and Suggestions for Authors

Round 2

Reviewer 1 Report

Comments and Suggestions for Authors

It can be accepted in this version.

Reviewer 2 Report

Comments and Suggestions for Authors

The present version is enough good